

# Weichselian permafrost depth in lowland Europe: a comprehensive uncertainty and sensitivity analysis

Joan Govaerts[1], Koen Beerten[1], Johan ten Veen[2]

[1] SCK•CEN, Institute Environment-Health-Safety, Boeretang 200, 2400 Mol, Belgium
[2] TNO Geological Survey of the Netherlands, Princetonlaan 6, 35584 CB Utrecht, The Netherlands

*Correspondence to*: Joan Govaerts (jgovaert@sckcen.be)

Abstract. The Rupelian clay in the Netherlands is currently being the subject of a feasibility study with respect to the storage of radioactive waste in the Netherlands (OPERA-project). Many features need to be considered in the assessment of the long-term evolution of the natural environment surrounding a geological waste disposal facility. One of these is permafrost development as it may have an impact on various components of the disposal system, including the natural environment (hydrogeology), the natural barrier (clay) and engineered barrier. It thus seems crucial to determine how deep permafrost will develop, in order to properly address the possible impact on the various components. It is expected that periglacial conditions will reappear at some point during the next several hundred thousands of years, a typical timeframe considered in geological waste disposal feasibility studies. In this study, the Weichselian glaciation is used as an analogue for future permafrost development. Permafrost depth modelling using a best estimate temperature curve of the Weichselian indicates that permafrost would reach depths between 140 m and 180 m. Without imposing a climatic gradient over the country, deepest permafrost is expected in the south, due to the lower geothermal flux and higher average sand content of the post-Rupelian overburden. Accounting for various sources of uncertainty, such as type and impact of vegetation, snow cover, air surface temperate gradients across the country, possible errors in palaeoclimate reconstructions, porosity, lithology and geothermal flux, stochastic calculations point out that permafrost depth during the coldest stages of a glacial cycle such as the Weichselian, for any location in the Netherlands, would be between 120-200 m at the 2σ level. In any case, permafrost would not reach depths greater than 270 m. The most sensitive parameters in permafrost development are the mean annual air temperatures and porosity, while the geothermal flux is the crucial parameter in permafrost degradation once temperatures start rising again. The calculations presented here are robust and conservative.



# 1  Introduction

Northern hemisphere permafrost, or perenially frozen ground, is presently restricted to areas in close proximity to the Arctic, generally north of 60° North Latitude, and even further to the south in mountainous areas (e.g., Himalaya, northern Rocky Mountains). Today, the lowlands of northwestern Europe are located outside the cryosphere, as far as permafrost development and periglacial earth surface and deformation phenomena are concerned. However, palaeoclimatic reconstructions have pointed out that much of northwestern Europe experienced permafrost conditions during the maximum expansion of continental ice, termed the last glacial maximum (LGM; Huijzer and Vandenberghe, 1998), recently also termed the last permafrost maximum (LPM; Vandenberghe et al., 2014). Nothwithstanding the fact that recent studies suggest global warming to continue or accelerate during the next 100 years (Stocker et al., 2013), or that we might be facing an exceptionally long interglacial (Berger and Loutre, 2002), it is expected that periglacial conditions will reappear at some point during the next several hunderd thousands of years, a typical timeframe considered in geological waste disposal studies (BIOCLIM, 2001). Whereas the distribution, type and timing of frozen ground in the past is, generally speaking, relatively well known in northwestern Europe from the distribution of shallow subsoil periglacial deformation phenomena (Huijzer and Vandenberghe, 1998), the maximum depth of past permafrost development is difficult to observe in the geological record. Therefore, numerical simulation seems to be the most suitable tool to estimate past and future permafrost depths, and has been applied already in several case studies elsewhere in Europe (Deslisle, 1998; Grassmann et al., 2010; Kitover et al., 2013; Busby et al., 2015).

Many of the countries that experienced permafrost in the past, such as the Netherlands, Belgium, United Kingdom, France and Germany, are currently running feasibility studies for the disposal of radioactive waste in deep geological repositories (Grupa, 2014). The concept of geological disposal is based on a multi-barrier system. Engineered barriers, consisting of steel components, concrete and/or bentonite are designed to contain the nuclear waste. Subsequently, the engineered barrier is placed in a natural barrier consisting of a low-permeability host rock that is situated at sufficient depth and located in a stable geological environment. Permafrost development may have an impact on various components of the disposal system, including the natural environment, the natural barrier and engineered barrier. The hydrological cycle will completely change under permafrost conditions, up to the point where surface and subsurface hydrology become completely independent, and groundwater flow is drastically changed (Weert et al., 1997). Furthermore, landscape, vegetation and thus the biosphere will be completely different under permafrost conditions (Beerten et al., 2014). Hydraulic properties of aquifer sands and aquitard clays might be affected during permafrost, especially if accompanied by repeated freeze-thaw cycles (Othman and Benson, 1993; McCauley et al., 2002). Migration of radionuclides through the host rock may be altered and the mechanical properties of the engineered barriers may be affected due to freeze-thaw cycles (Busby et al., 2015). Pore-water chemistry might change due to outfreezing of salts, and gas-hydrates may develop (Stotler et al., 2009). Finally, microbial activity in the host rock will likely be affected. It thus seems crucial to determine how deep permafrost will develop, in order to



properly address the possible impact on the various components. Thick and deep low-permeability clay formations are often good candidates for host-rocks. The Rupelian clay, which is subcropping in most of the northwest European countries mentioned above (Vandenberghe and Mertens, 2013), is considered in the Dutch OPERA-project as a candidate host-rock (Grupa, 2014). In the present northwest European context it provides an interesting case study because it is distributed over a

wide range of settings with respect to overburden thickness, lithology, porosity and geothermal flux.

Usually, uncertainties in permafrost modelling studies in the context of radioactive waste disposal are not addressed in a systematic way (e.g., Busby et al., 2015, Holmen et al., 2011, Hartikainen et al., 2010 ). The sensitivity of permafrost models is often assessed in terms of variations to single parameters like porosity or surface temperature, each one at a time (Kitover

et al., 2013). These so-called local Sensitivity Analyses (SA) assess the response of the model output to a small perturbation of single parameters, one at a time, around a nominal value. The main disadvantage of this method is that information about the sensitivity is only valid for this very specific location in the parameter space, which is usually not representative of the physically possible parameter space, and becomes problematic especially in the case of non-linear models. To overcome this problem, a global stochastic sensitivity analysis is used in this work, where multiple locations in the physically possible

parameter space are evaluated at the same time.

In this paper, we will present permafrost depth calculations for a number of areas in the Netherlands that are representative for a specific geological setting using a best estimate temperature curve for a glacial cycle that is analogues to the last ice age (Weichselian). In addition, results from stochastic simulations will be presented that give an indication of the probability of permafrost depths under a future glacial climate, taking into account various combinations of temperature, overburden

lithology, porosity and geothermal flux. Furthermore, a sensitivity analysis is performed to identify the key model parameters that cause the uncertainty on the calculated permafrost depths.

The results of the simulations can be used to assess if the foreseen depth of a future geological disposal facility in the Netherlands is sufficient to exclude possible malicious effects from the presence of developing permafrost. Up to the present, these kind of simulations were not performed nationwide, and the thorough uncertainty and sensitivity analysis adds to the

robustness of the findings.





## 2  Mathematical and numerical model for permafrost growth and degradation

To describe heat transport in the subsoil of the Netherlands, the following one-dimensional enthalpy conservation equation is used with heat transport only occurring by conduction.

$$C_{eq}\frac{\partial T}{\partial t} + \nabla \cdot (-\lambda_{eq}\nabla T) = Q \qquad (1)$$

where $C_{eq}$ is the effective volumetric heat capacity (J/K·m³), $T$ is temperature (K), $\lambda_{eq}$ is the effective thermal conductivity (W/m·K), and $Q$ is a heat source (W/m³).

When modelling the thermal effects of freezing and thawing, equation (1) has to include three phases: rock matrix, fluid and ice. To achieve this, the following volume fractions are defined:

$$\theta_m = 1 - \theta, \theta_f = \theta \cdot \Theta, \theta_i = \theta - \theta_f \qquad (2)$$

The subscripts $f$, $i$ and $m$ account for the mixture between solid rock matrix ($m$), fluid-filled pore space ($f$) and ice filled pore space ($i$). This mixture is characterized by porosity $\theta$ and $\Theta$ denotes the fraction of pore space occupied by fluid. As a result of the complicated processes in the porous medium, melting cannot be considered as a simple discontinuity. $\Theta$ is generally assumed to be a continuous function of temperature in a specified interval.

When a material changes phase, for instance from solid to liquid, energy is added to the solid. This energy is the latent heat of phase change. Instead of creating a temperature rise, the energy alters the material's molecular structure. This latent heat of freezing/melting of water, $L$, is 333.6 kJ/kg (Mottaghy & Rath, 2006). $C_{eq}$ is a volume average, which also accounts for the latent heat of fusion:

$$C_{eq} = \theta_m \rho_m c_m + \theta_f \rho_f \left(c_f + \frac{\partial \Theta}{\partial T}L\right) + \theta_i \rho_i \left(c_i + \frac{\partial \Theta}{\partial T}L\right) \qquad (3)$$

where $\theta$ is the volumetric content, $\rho$ equals density (kg/m³), and $c$ is the specific heat capacity (J/(K·kg)). It includes additional energy sources and sinks due to freezing/melting using the latent heat of fusion $L$ for only the normalized pulse around a temperature transition $\frac{\partial \Theta}{\partial T}$ (K$^{-1}$). The integral of $\frac{\partial \Theta}{\partial T}$ must equal unity to satisfy the condition that pulse width denotes the range between the liquidus and solidus temperatures. During heating, solidus is that temperature at which a solid begins to melt. Between the solidus and liquidus temperatures, there will be a mixture of solid and liquid phases. Just above the solidus temperature, the mixture will be mostly solid with some liquid phases. Just below the liquidus temperature, the mixture will be mostly liquid with some solid phases. This approach is similar to the one used by Mottaghy & Rath (2006),



Bense et al. (2009), Noetzli & Gruber (2009), Holmén et al. (2011) and Kitover et al. (2013). Values for the porosity, density and specific heat of the different components are given in Table 1.

The values of the specific heat of the Boom Clay matrix are obtained from Cheng et al. (2010). The equivalent heat capacity
5    then adds up to 1443 J/(kg K) and 981 J/(kg K) for the fully unfrozen and frozen state respectively. This is in the same range as the values used in Marivoet & Bonne (1988) and Kömle (2007) for clay sediments. The value of the sand matrix is set to a value inside the ranges which are found for quartz minerals and sands (see Mallants (2006) and references therein). For sandy soils, the equivalent heat capacity adds up to 1319 J/(kg K) and 937 J/(kg K) for the fully unfrozen and frozen state respectively when a porosity of 30% is assumed.

In case of a phase change at a single temperature, thermal conductivity is not continuous with respect to temperature. However, considering the freezing range in rocks, we use equation (2) and (4) for taking into account the contributions of the fluid and the ice phase. Since the materials are assumed to be randomly distributed, the weighting between them is realized by the square-root mean, which is believed to have a greater physical basis than the geometric mean (Mottaghy & Rath,
2006).

$$\lambda_{eq} = \left( \theta_m \sqrt{\lambda_m} + \theta_f \sqrt{\lambda_f} + \theta_i \sqrt{\lambda_i} \right)^2 \qquad ( 4 )$$

Values for the thermal conductivity of the different components are given in Table 1.

The values of the thermal conductivity of the rock matrix of Boom clay and sand are chosen in the same order of magnitude
of the values used by Bense et al. (2009) and Mottaghy & Rath (2006), who used respectively 4.0 W/(m K) and 2.9 W/(m K) for a generic sediment rock species.
For Boom clay, the equivalent thermal conductivity then adds up to 1.31 J/(kg K) and 2.03 J/(kg K) for fully unfrozen and frozen state respectively. The conductivity value of unfrozen Boom Clay is thus equal to the vertical conductivity obtained from the ATLAS 3 study (Cheng et al., 2010).
In sandy soils, the equivalent thermal conductivity is 2.05 W/(m K) and 2.80 W/(m K) for the fully unfrozen and frozen state respectively. The values are in the same range as the values found in Mallants (2006) and references therein.
The heat transport equation is implemented in COMSOL multiphysics, Earth science Module (2008), together with all the correlations for the thermal properties. Because the thermal properties differ between the frozen and unfrozen state, a variable Θ is created, which goes from unity to zero for fully unfrozen to frozen. Therefore, the effective properties switch
with the phase through multiplication with Θ. The switch in Θ from 0 to 1 occurs over the liquid-to-solid interval (0.0°C to -0.5°C) using a smoothed Heaviside function. The dependency of the freezing point of water on pressure, salinity and the adsorptive and capillary properties of the soil is conservatively neglected in the present calculations.



As such, the freezing process is modelled using a gradual and not a sudden uptake/release of the latent energy, starting at 0°C to ensure numerical stability. Ideally, this freezing interval should be kept as small as possible. However, decreasing the size of the interval will increase the computational burden and compromise the numerical stability. The effect of the size of the freezing interval on the permafrost depth was previously investigated by Govaerts et al. (2011, Appendix A). It was shown that there is little influence on the resulting temperature profiles as the 50% frozen isoline (which corresponds to the center temperature value of the liquid-solid interval) is nearly identical for all the values that have been tested in that study. The size of the liquid-to-solid temperature interval does not seem to impact the retardation of the cold wave, also known as the zero curtain effect. However the positions of the 0 and 100 % frozen isolines are severely sensitive to this width, and the exact value is uncertain as it can range from 0.5 to 2 °C depending on the material type (Noetzli & Gruber, 2009). Therefore, the choice for the -0.25°C as permafrost indicator was made, as we want to present our results in a robust manner, independent from modelling assumptions.

Due to the 1D nature of the model, mesh size poses no strong obstacle. The results are checked for grid-independency and as a result a mesh of 500 elements is chosen as an optimal setting for the final simulations.

## 2.1 Parameters, initial and boundary conditions used in the reference calculation

Permafrost development is dependent on atmospheric and surface boundary conditions, basically temperature and vegetation, and subsurface properties such as lithology, porosity and geothermal flux. As such, it is the result of interactions between global changes (temperature) and local conditions (subsurface geology). The strategy adopted for this specific study consists of the following elements. First, we try to simulate a future glacial climate using the Weichselian glaciation (115-11 ka) as an analogue. Various temperature estimates are available for this glacial period, many of them being derived from palaeoclimatological archives in Belgium and the Netherlands. The input temperature is held constant for the entire country. Subsequently it will be used to force the subsurface permafrost model, which is fragmented into different representative polygons. The initial condition of the model is the steady state temperature profile based on the present day temperature gradient.

The thickness of subsurface units and their lithofacies distribution are considered relevant for permafrost modelling as this affects porosity and the effective thermal properties. The selection of the different areas for permafrost modelling is based on the presence of 17 structural elements, including 6 highs, 5 basins and 6 platforms (Fig. 1). The rationale is that these structural elements delineate differences in thickness- and depths of both Mesozoic and Cenozoic subsurface units. Subsequently, a geological (property) model was constructed based on the surfaces of the DGM shallow subsurface model. For each unit, vertical gridcells with a height equal to unit thickness of 250x250m were constructed. These gridcells were populated with the parameters described before. Subsequently, all parameters were averaged over the vertical interval



overlying the Rupel Fm. (the overburden). The research area is subdivided into several polygons which dimensions range roughly between 9x15km and 110x140km. The midpoint positions of the various polygons are plotted in Fig. 1.

The reference calculation consists of 17 simulations of permafrost pro- and degradation during the last interglacial-glacial-interglacial cycle. Each calculation is performed for one of the seventeen polygons. These are discussed in more detail in the following section. It should be noted that this method does not allow for local variations to be included, but rather serves to highlight regional trends over the Netherlands.

## 2.2    Porosity, lithology and overburden thickness

An important parameter for permafrost modelling is porosity. Porosity is directly linked with water content and thus thermal properties of the soil. (see Table 1). A porosity value is assigned to each of the lithostratigraphic units defined in the Digital Geological Model (DGM; Gunnink et al., 2013) of the shallow Dutch subsurface. The hydrogeological model REGIS provides a further subdivision and includes both aquifers (sand) and non-aquifer layers (clay). Using the REGIS information a percentage of clay vs. sand for each of the DGM units can be calculated. Based on a relatively small amount of porosity measurements of the sand and clay layers in the stratigraphic interval above the Rupel Fm., a best fit, generally applicable, porosity-depth relationship was established for all sandy and clayey depositional facies of the various units. This allows to make porosity predictions in non-studied domains.

Several trends can be observed from Fig.2. The highest averaged mid-depth porosities for the post-Rupelian overburden are observed in the east and the southwest, reaching 50%. The lowest values are found in the southeast (Roer Valley Graben, polygon RVG) and the northwest. The porosity is basically influenced by two other parameters: lithology and burial depth. The thicker the post-Rupelian overburden, the deeper the mid-depth for which the porosity is determined, and thus the lower the porosity. Lithology also has an influence on porosity because on average, clay has a higher porosity than sand. As such, the porosity map is a mirror image of a combination of lithology and overburden thickness. Note that the depth to the top Rupel Fm. is representative for the total overburden depth, which is strongly coupled to the tectonic setting, i.e., thick in basins and grabens and relatively thin on structural highs (Fig. 1 and Fig. 2). Lithofacies (given as % of sand) seems to be linked to certain structural elements. Notably in the southeast of the Netherlands, the sand content is very high, reaching more than 80% in the RVG and adjacent areas. These areas acted as traps for a thick series of Neogene continental sands.

The total vertical length of the one-dimensional lithological domain is extended to at least 500 m with clay, in case the overburden does not reach this depth. This implies assuming that the Rupelian layers underlying the overburden are sufficiently thick to bridge the distance from the bottom of the overburden to a depth of 500 m. This is needed as the lower boundary condition needs to be imposed at a distance sufficiently far from the top to avoid artificial, numerical interaction with the surface temperature boundary condition.





## 2.3    Upper boundary condition: temperature evolution of a future glacial cycle

The temperature curves and data used in this study are shown in Fig. 3. Best estimates for the mean annual air temperature (MAAT) during MIS5 (marine isotope stage 5) is based on pollen data from van Gijssel (1995), but replotted against a more
recent chronostratigraphical framework for the Weichselian glaciation (see e.g., Busschers et al., 2007). The main features of the MIS5 climate are the relatively mild stadials 5b and 5d, with an MAAT of -2°C, and the relatively cold interstadials 5c and 5a, with an MAAT of +4°C. The first period with continuous permafrost development in the Netherlands would have been MIS4, with MAAT values dropping to as low as -4°C (threshold for discontinuous permaforst) and even -8°C (threshold for continuous permafrost) for the end of MIS4. These values are based on periglacial deformation phenomena
(e.g., ice-wedge casts and large-scale involutions) in the shallow subsoil and their present-day distribution in areas of stable continuous permafrost (Vandenberghe and Pissart, 1993; Huijzer and Vandenberghe, 1998; Vandenberghe et al., 2014). The following MIS3 is characterised by a somewhat milder climate, showing less periglacial deformation of the subsoil. Analysis of flora and fauna preserved within MIS3 sediments, and the type and nature of periglacial deformation shows that some interstadials might have reached an MAAT between 0°C and +6°C (e.g., Upton Warren, Hengelo and Denekamp
interstadials; Huijzer and Vandenberghe, 1998, Busschers et al., 2007 and van Gijssel, 1995), and several stadials would have reached an MAAT as low as -4°C (e.g., Hasselo stadial; Busschers et al., 2007). Subsequently, the climate evolves towards the Late Glacial Maximum (LGM), which is situated in MIS2. Data for this stage is mainly derived from Renssen and Vandenberghe (2003) and Buylaert et al. (2008), and is based on the presence and type of periglacial deformation phenomena. The MAAT for the period between 28 ka and 15 ka would not have exceeded 0°C, while some periods show
MAAT values as low as -8°C. The end of MIS2 is characterised by a stepwise trend towards global warming, showing significant variations between interstadials (Alleröd) and stadials (Younger Dryas). Finally, present-day MAAT values of around +10°C are attributed to MIS1.

Upper and lower bounds for these temperature data are given in Fig. 3. They serve as input for the permafrost depth uncertainty analysis. Instead of using one best estimate temperature evolution for the Weichselian glaciation used as an
analogue, a minimum and maximum temperature distribution for each timeframe is determined, which will be randomly sampled to produce various combinations of upper and lower bound MAAT values in the stochastic uncertainty analyses. Different sources of uncertainty are thus taken into account, such as the reliability of the palaeotemperature proxy, the transferability towards a future glacial climatic cycle, temperature gradients across the country and atmosphere-soil temperature coupling. Lower bound estimates are set ca. 3-4°C lower than the best estimate. This range allows for
uncertainty with respect to the palaeotemperature proxy in the case of discontinuous permafrost, which might exist up to MAAT values as low as -8°C before it turns into continuous permafrost (Huijzer and Vandenberghe, 1998). For the coldest periods, such as the LGM, the lower bound MAAT was set to -11°, or 21°C below the present day MAAT. This allows including the possibility of a colder future glacial than the Weichselian analogue would suggest (e.g., the Saalian, when the



Fennoscandinavian ice-sheet margin was reaching the Netherlands; Svendsen et al., 2004). Furthermore, a 21°C temperature drop for the LGM is in line with the coldest estimate reconstructions for low lying areas in Great Britain (Busby et al., 2015). As a result, the MAAT remains below 0°C throughout much of the glacial cycle, except for several short interstadials, such as the Alleröd.

The upper bound is based on a warm reconstruction of the Weichselian climate, which is based on a pollen sequence in sediments from the crater lake at La Grande Pile, situated ca. 500 km south of Amsterdam at an altitude of ca. 350 m (Guiot et al., 1989). The mean MAAT estimate obtained in that study is used here as an absolute maximum scenario for the Weichselian glaciation in northwestern Europe given its location. The upper bound for the time period around 20 ka, for which the pollen record gives no solution, is set to -4°C (or 14°C below the present-day MAAT), because we assume that at

least discontinuous permafrost would have developed in northwestern Europe around that time period. This value is slightly lower than globally reconstructed temperatures for the LGM in northwestern Europe changes at the LGM for northwestern Europe, as proposed by Annan and Hargreaves (2013) using climate modelling and proxy data.

The temperature data presented here are directly used as soil input data. However, vegetation, snow or ice would act as a shield against penetration of cold air and hamper the evolution of permafrost with depth. The effect of vegetation and snow

can be addressed using the concept of freezing and thawing factors, which are equal to one in the case of no vegetation or snow and gradually decrease with vegetation type and snow thickness (Lunardini, 1978). During a Weichselian glacial cycle as the one presented here, the shielding effect of snow and temperature would increase the mean annual soil surface temperature (MAST) with 2°C to 5°C with respect to the MAAT (northern Belgium; Govaerts et al., 2011). Departing from the beste estimate temperature curve, this range is almost entirely covered by the uncertainty range (Fig. 3).

Finally, it has to be mentioned that the model is conservative with respect to the following phenomena. Firstly, vadose zone hydrology is neglected. During very cold stadials, infiltration would probably be so low that the groundwater table would be significantly lower. Unsaturated soil slows down permafrost development because of the difference in thermal conductivity between air and water. Next, groundwater flow is neglected but it is evident that this would slow down the speed of permafrost development as well because of the redistribution of heat. Finally, outfreezing of pore water salt would lower the

speed of permafrost development because more latent heat is needed to freeze water with elevated salt concentrations.

## 2.4     Lower boundary condition: geothermal flux

For each of the 17 polygons, the mid-depth temperature and the surface temperature were used to calculate the temperature

gradient. These data are then used to calculate the geothermal flux using the average the thermal conductivity values (Fig. 2D). Generally speaking, the country is split up between a southeastern part with lower values of the geothermal flux (0.06-0.07 W/m²) and a northwestern part with higher flux values (up to 0.09 W/m²). It is interesting to note that this pattern follows the pattern of recent differential tectonic land movement as calculated by Kooi et al. (1998). The higher flux values





in the northwest can also be linked to higher mid-depth temperatures in that area. These in turn are the result of the presence of Zechstein salt layers that have a relatively high thermal conductivity (ca. 4 W/(m.K); Bonté et al., 2012).

## 2.5 Stochastic uncertainty and sensitivity analysis

### 2.5.1 Uncertainty analysis

The goal of a safety case for a final repository project is to prove that the facility will be safe in every aspect. This comprises considerations about the near and far future. It is a principal fact, however, that statements about the future can never be more than likelyhood statements. Although, by this reason, a strong proof of safety is principally impossible, the remaining uncertainty can be assessed and should be kept as small as possible. This has to be done by carefully identifying and

10 quantifying the primary uncertainties that can have an influence on the overall uncertainty of the safety statement and properly assessing this influence.

As an integral part of a safety case file, supporting calculations for radioactive waste disposal often involves the analysis of complex systems. Various types of uncertainty affect the results of the evaluations. An overview of the treatment of uncertainties in the disposal programmes of several European countries has been compiled within the PAMINA project

(Marivoet et al., 2008).

The nature of the uncertainty can be stochastic (or aleatory) or subjective (or epistemic). Epistemic uncertainty derives from a lack of knowledge about the adequate value for a parameter/input/quantity that is assumed to be constant throughout model analysis. In contrast, a stochastic model will not produce the same output when repeated with the same inputs because of inherent randomness in the behavior of the system. This type of uncertainty is termed aleatory or stochastic.

In general a distinction is made between three sources of uncertainty:uncertainty in scenario descriptions, including the evolution of the main components of the repository system; uncertainty in conceptual models;
uncertainty in parameter values.

Although both types and even sources of uncertainties cannot be entirely separated, the work in this report deals mostly with subjective (or epistemic) uncertainties which are reflected in the uncertainties in parameter values.

Parameters, initial and boundary conditions for mathematical models are not very often known with a high degree of certainty. The study of parameter uncertainty is usually subdivided into two closely related activities referred to here as uncertainty analysis and sensitivity analysis, where (i) uncertainty analysis (UA) involves the determination of the uncertainty in analysis results that derives from uncertainty in input parameters and (ii) sensitivity analysis (SA) involves the determination of relationships between the uncertainty in analysis results and the uncertainty in individual analysis input

parameters. SA identifies the parameters for which the greatest reduction in uncertainty or variation in model output can be obtained if the correct value of this parameter could be determined more precisely.



To this end, a Monte Carlo simulation is based on performing multiple model evaluations using random or pseudo-random numbers to sample from probability distributions of model inputs. The results of these evaluations can be used to both determine the uncertainty in model output and perform SA. The popular and robust Monte Carlo (MC) method in combination with the efficient Latin Hypercube Sampling (LHS) is used here, and is described exhaustively in several

review papers and textbooks (e.g. Marino et. al, 2008; Helton, 1993). LHS requires fewer samples than simple random sampling and achieves the same level of accuracy.

### 2.5.2 Implementation

The calculations are done in three steps, using a Matlab 2012a (The Mathworks) linked to the finite element PDE solver Comsol 3.5a (2008). First, values of all selected stochastic input variables are sampled for all the runs using in-built Matlab functions. All variables are assumed to be independent so no in-between correlations were implemented. A number of simulations are then performed using the sampled parameter combinations. Matlab was used to automate the simulations performed by the FE code COMSOL for all Monte Carlo runs. The tables of collected results are produced by Matlab can

then be directly analysed with Matlab to calculate and plot the percentiles and mean value of the permafrost depth as a function of time. Then again Matlab was used to compute e.g. Standardized or Partial Correlation Coefficients to investigate the parameter sensitivity. For the regression based analyses, 1000 realizations are performed to obtain the results for each scenario. In order to guaranty stability of the output, enough number of realizations should be provided. The minimum number of realizations required to assure stable output depends on the system itself and the number of uncertain variables

associated with it. In this work, a number of 1000 realisations were performed. Helton (2005) showed that 100-300 model runs were sufficient for stable results using a far more complex two-phase flow model with 37 uncertain variables.

### 2.5.3 Parameter Ranges

The stochastic simulations will give an indication of the probability of nation-wide permafrost depths under a future glacial climate, taking into account various combinations of temperature, overburden lithology, porosity and geothermal flux.

A proper quantification of uncertainties in the form of probability density functions (pdfs) is an essential part of the uncertainty management and a pre-requisite for probabilistic uncertainty and sensitivity analysis. As the actual knowledge

about the statistical distribution of the parameters in question is limited, it is only possible to estimate a minimum, a maximum and a most probable value. In this case the tri-angular distribution is the most appropriate expression of the state of the knowledge (Bolado et al., 2009).



The parameters that are investigated in the stochastic analysis are shown in Table 2. . Their minimum, maximum en mode values are used to build a tri-angular probability density functions which are sampled in the stochastic analysis. T1 to T26 are variables which are used to control the magnitude of the various temperature plateaus during the Weichselian temperature cycle. This allows to account for the actual parameter uncertainty on the temperature as well as the nation-wide spatial parameter variability.

### 2.5.4    Sensitivity analysis

A wide range of SA methods exists but can generally be classified into Local and Global techniques.  Local SA will be assessing the response of the model output to a small perturbation of single parameters (the so called one at a time method) around a nominal value.  The main disadvantage of this method (and other local SA methods) is that information about the sensitivity is only valid for this very specific location in the parameter space only, which is usually not representative of the physically possible parameter space, which becomes problematic especially in the case of non-linear models.

To deal with this problem global SA methods have been developed, where multiple locations in the physically possible parameter space are evaluated at the same time. The most frequently used global techniques are implemented using Monte Carlo simulations and are therefore called sampling-based methods. Global SA with regression-based methods rests on the estimation of linear models between parameters and model output. For linear trends, linear relationship measures that work well are the Pearson correlation coefficient (CC), partial correlation coefficients (PCCs), and standardized regression coefficients (SRC) (Helton, 2005). In this study, the SRC will be used.

## 3    Results

### 3.1    Permafrost development during a Weichselian temperature cycle – reference case

The definition of permafrost applied here is ground which remains at or below 0 °C for at least 2 consecutive years  (French, 2007). The -0.25°C isotherm is chosen as a conservative estimate of the permafrost front as explained earlier. Due to the choice of a gradual phase change between 0 °C and -0.5 °C, a soil at -0.25°C corresponds to a mixture of 50% water and 50% ice.

The progradation fronts (= location where the temperature reaches 0°C,-0.25 °C and -0.5°C the soil is resp. 0%, 50% and 100% frozen) for the FRP and LBH polygons are shown in function of time (Fig. 4).  The permafrost front penetrates about





150m to 180m into the subsoil depending on the location, as a result of extremely low mean annual air temperatures during the final phase of MIS4 (early Pleniglacial) and the middle part of MIS2 (late Pleniglacial)..

The spatial distribution of maximum permafrost depth at any time during a Weichselian climatic analogue is given in Fig. 5. The maps are interpolated (inverse distance weighted) from individual polygon results, and are the result of model forcing by

the best estimate climate evolution given in Fig. 3. The maximum permafrost depth generally corresponds with the coldest peak in MIS2 (around 20 ka BP). The depth of the location where 50% of the pore water is frozen, ranges from 140m to 180m. The spatial variability in permafrost depth is further illustrated along a N-S transect, from polygon centre FRP in the north to LBH in the south (Fig. 6).

Somewhat surprisingly, the calculated permafrost depth would be about 40 m less in the north. Intuitively, one would expect

permafrost to reach greater depths in the north, because of the inferred temperature gradient over the country. As stated above, the input temperature was kept constant for the entire study area, such that the results can be interpreted solely in terms of subsurface properties. The spatial pattern of maximum permafrost depth is in fair agreement with the pattern of geothermal flux, as shown in Fig. 2, and a relationship with the weight fraction of sand can be observed as well. This seems logical as a higher geothermal flux imposes a stronger resistance against the intrusion of subzero temperatures into the soil.

A higher sand fraction facilitates permafrost growth, as a sand matrix has a higher thermal conductivity which allows a more rapid extraction of thermal energy towards the surface during cold periods.

Thus, assuming a constant temperature evolution over the Netherlands, geothermal fluxes, and to a lesser extent sand percentage, seem to be the determining factor to explain the N-S variability of the maximum permafrost depth. Parameter sensitivity will be addressed in more detail in the section on the sensitivity analysis.

## 3.2    Results of the uncertainty analysis

The uncertainty analysis translates the uncertainty on the input parameters into an uncertainty on the permafrost depth (= 50% frozen isoline). The results are shown in Fig. 7. The median values of the maximum permafrost depth at a time of 20 ka

are about 150 m, while the most conservative parameter combinations result in permafrost fronts going as deep as 270 m. Note that maximum permafrost depth for the 95-100 percentiles occurs after the thermal minimum for the cold phase around 60 ka BP, and not during the LGM (20 ka BP). This is caused by a number of simulations in which the random combination of cold temperatures in the period of 80 to 60 ka BP result in a very long and continuously cold period with temperatures ranging from -4°C to -11°C. The difference between the 5% and 95%-percentiles (2σ) is about 80 m, which is quite high

given the relatively low number of parameters.




### 3.3 Results of the sensitivity analysis

The goal of this sensitivity analysis (SA) is to determine the relationships between the uncertainty in output and the uncertainty in individual input parameters. SA identifies the parameters for which the greatest reduction in uncertainty or variation in model output can be obtained if the correct value of this parameter could be determined more precisely. The results are analysed by looking at the evolution of the SRC and PCC coefficients. PCC and SRC provide related, but not identical, measures of the variable importance. If input factors are independent, PCC and SRC give the same ranking of variable importance. This is the case in the present study, and we will focus on the SRC to discuss the sensitivity analysis. A positive correlation coefficient (SRC/PCC) means that a higher value of the parameter will cause a larger permafrost depth and vice versa.

It can be seen in Fig. 8 that the $R^2$-values are close to 1, which indicates that the regression model is accounting for most of the uncertainty in the permafrost depth, and the model is behaving in a linear way.

The SRC indicates that the geothermal flux is the most important parameter, besides temperature forcing. It is interesting to note that during permafrost growth (e.g. around 90 ka), the geothermal flux is equally important as the porosity. However, when the surface temperature again rises and the permafrost starts to degrade, the geothermal flux acts as the main driving force of the melting process.

During the course of simulation time, correlation coefficients can change their sign. During permafrost growth, at the initial phase of a subzero temperature period, a higher porosity will hamper permafrost growth. A larger pore water content means that a larger amount of energy needs to be removed from the subsoil in order to cool it down, because of the larger effective heat capacity, and to induce a phase change of the total amount of pore water. A larger water content also decreases the total effective thermal conductivity which slows down the extraction of thermal energy towards the surface. Thereafter, during the subsequent warmer period, a higher ice content will require a larger amount of heat to be supplied to melt away the permafrost.

The sand fraction shows a relatively strong, positive influence on permafrost depth, this confirms the findings of the nationwide simulation. Compared to clay, sand has a higher thermal conductivity which will cause a more rapid cooling of the subsurface during cold periods.

The overburden thickness only seems to play a role during moderately cold periods (e.g. MIS 5b and 5d). If the overburden thickness is lower than 500m, the remaining part of the domain is repleted with Boom Clay type material, which has a slightly lower thermal conductivity than the often sandy overburden, which slows down permafrost formation. This only seems to be of any importance in periods when the surface temperature is only slightly below the freezing point. Later, when the cold periods become more extreme (e.g. MIS 2), the difference in thermal conductivity at the clay/sand interface plays a less prominent role, and the other parameters become more significant.



Finally, it is no surprise that, being the driving force for the formation of a permafrost, the surface temperature is crucial at the time it is imposed at the top of the computational domain (Fig. 8: in order not to overcrowd the figure, only 8 of the 26 temperature variables are shown). It is important to note that a specific correlation coefficient becomes larger when that temperature is maintained for a longer period (e.g., T2 and T4). Closer to the present, the dynamics of the temperature evolution during the Weichselian are better captured in the proxy data, which translates itself to a more detailed temperature evolution during the last 50 ka (T12-T26). Subsequently, this makes the individual temperature parameters seem less important compared to the earlier temperatures, which can be seen as an artefact induced by the dynamics of the temperature curve.

Another interesting point to note is the fact that a cold temperature during an early timeframe can still manifest its influence thousands of years later. For instance, the PCC-curves of T6 and T7 show a long tailing which still impacts the formation of the permafrost around 60 ka. This can be explained by the thermal inertia of the frozen soil, which has not been fully reverted to the initial temperatures at the start of a subsequent cold period.

## 4 Discussion

The best estimate permafrost depth values of 140-180 m (50% ice and 50% water) calculated here for the Weichselian glaciation in the Netherlands are somewhat lower than the 200 m that was obtained by Govaerts et al. (2011) using the same model and the same temperature curve. In the latter study, a lower porosity was used (down to 30%), which favours the development of permafrost and thus explains the deeper permafrost. We now compare the results to other permafrost depth modelling results for NW Europe during the Weichselian or a future analogue from Delisle (1998), Grassmann et al. (2010), Hartikainen et al. (2010), Holmén et al. (2011), Kitover et al. (2013) and Busby et al. (2015). These different modelling exercises revealed quite contrasting permafrost depths for the LGM, which is not surprising given different approaches and parameter values were used with respect to palaeotemperatures, duration of cold phases, subsoil thermal properties, porosity, heat flux, ice advance, etc. The values derived from the different studies range between ca. 100 m and ca. 300 m for a western European context, being site or non-site specific. Kitover et al. (2013) used an extended cold period of unknown duration and a MAAT of -8°C to produce steady-state continuous permafrost, reaching 300 m depth. Holmén et al. (2011) calculated a median permafrost depth for northwestern France of ca. 120 m using a MAAT of -6°C for the coldest peak during a future Weichselian analogue. The value of 100 m from Delisle et al. (1998) is based on a relatively high MAAT of -7°C for the LGM during a very small time period at around 18 ka. Furthermore, that study used a fairly high geothermal heat flux, 60 mW/m², as is the case for the study by Govaerts et al. (2011). The latter however used a lower MAAT (-9°C) for the LGM, persisting for 2000 years and preceded by already very cold temperatures in the millennia before. Similar permafrost depths were obtained in the study by Grassman et al. (2010) for northern Germany covering the last 1 Ma. The maximum



depth of 300 m for the permafrost base was realised during a cold peak around 400-450 ka with a mean annual ground surface temperature of -10°C and a heat flux of 50 mW/m². Busby et al. (2015) calculated that for an extreme cold climate, based on the last glacial-interglacial cycle in Great Britain, maximum permafrost would vary between 180 m (South Midlands) and 305 m (Southern Uplands). The largest value of 305 m was obtained using an MAAT of ca. -20°C for the coldest peak, which is about 8°C lower than the one used in the present study. Note that in the coldest Southern Uplands scenario ice-sheet advance was considered during the glacial cycle, which has a shielding effect on atmosphere-soil coupling. However, as only ground temperature has been modelled by Busby et al. (2015), they could not make any statements about ice formation or the depth that partially or completely frozen ground might penetrate to. Hartikainen et al. (2010) also defined several permafrost cases with maximum modelled permafrost depths (0 °C isotherm) of 260 m and 390 m respectively for the Forsmark site in Sweden.

The stochastic approach applied in this study combines a large range of parameter values and boundary conditions. Interestingly, this results in a permafrost depth distribution ranging between 100 m and 270 m (entire distribution for any time during the Weichselian glaciation). This window seems to cover any of the calculated permafrost depths as mentioned by previous studies. Some extreme values for the British Isles and Sweden that go beyond the thickest permafrost calculated in the present study are basically caused by much lower MAAT values for the coldest glacial peaks. Indeed, one of the remaining uncertainties are the duration and minimum MAAT values adopted for these cold peaks. Temperature reconstructions, based on periglacial deformation phenomena, show that the MAAT during the LGM in the Netherlands would have been equal to or lower than -8°C. In our coldest scenario, we adopted a value of -11°C that would last for about 2000 years. To some extent these minimum scenario values remain somewhat arbitrarely, and the question is how much they should be lowered in order to cover any possible future glacial scenario. Notwithstanding these remaining uncertainties, so far the 300 m depth may be regarded as a maximum value, because the model is conservative with respect to vegetation and snow cover, groundwater flow and the depth of the unsaturated zone. These contributing factors would all tune permafrost towards shallower depths. Future research should focus on defining an absolute minimum value for the MAAT used in permafrost calculations, based on natural analogues or a quantitative evaluation of periglacial deformation phenomena in the shallow subsoil through thermo-hydro-mechanical modelling of ice-wedge casts. It is also noted that the geothermal gradient, as used here for geothermal flux calculation, might not be in equilibrium yet with present-day climatic conditions (ter Voorde et al., 2014). This should be taken into account in future permafrost modelling work.

## 5    Conclusions

Permafrost depth modelling using a best estimate temperature curve of the Weichselian as an analogue for the future indicates that the permafrost front (50% ice and 50% water) would indicate permafrost depths between 140-180 m in the Netherlands. Using the same climatic data for the entire country, deepest permafrost is expected in the south, due to the



lower geothermal flux and higher average sand content of the post-Rupelian overburden. Taking into account various sources of uncertainty, such as type and impact of vegetation, snow, air surface temperate gradients across the country, possible errors in palaeoclimate reconstructions, porosity, lithology and geothermal flux, stochastic calculations point out that permafrost depth during the coldest stages of a glacial cycle such as the Weichselian, for any location in the Netherlands, would be between 120-200 m at the $2\sigma$ level. In any case, permafrost would not reach depths greater than 270 m. The most sensitive parameters in permafrost development are the mean annual air temperatures and porosity, while the geothermal flux is the crucial parameter in permafrost degradation once temperatures start rising again. The permafrost depth distribution presented here encompasses many of the previously calculated values for other contexts. Furthermore, the uncertainty analysis in which a wide range of parameter values are considered makes the results presented here directly relevant for any western European lowland location with a sedimentary overburden.

The calculations presented here are robust and conservative. However, in order to further reduce existing uncertainties, the following issues should be addressed: the effect of groundwater flow, salt content, snow and vegetation cover and an MAAT gradient over the country.

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



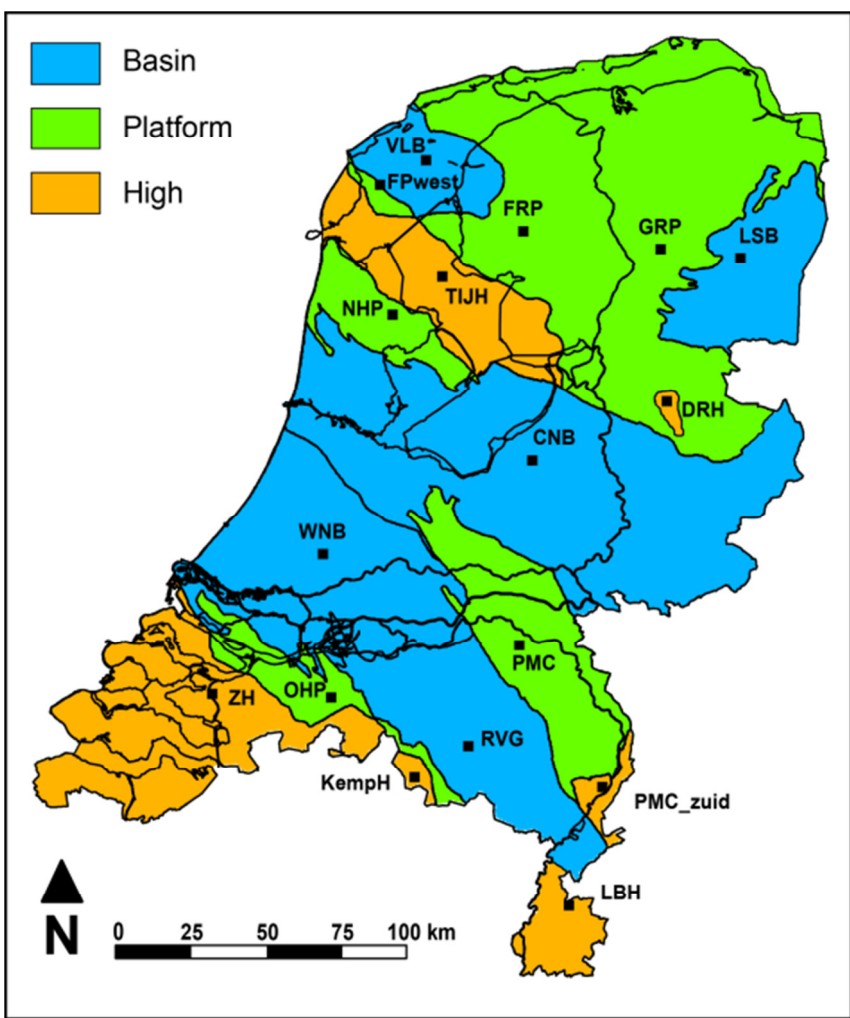

Figure 1 – Structural elements in the Durch subsurface. LBH = Limburg High, RVG = Ruhr Valley Graben, KempH = Kempisch High, ZH = Zeeland High, OHP = Oosterhout Platform, PMC_zuid = Peel-Maasbommel High, WNB = West

5 Netherlands Basin, CB = Central Netherlands Basin, DRH = Drenthe High, NHP = Noord Holland Platform, TIJH = Texel-IJsselmeer High, FP (west) = Friesland Platform, VLB = Vlieland Basin, GRP = Groningen Platform, LSB = Lower Saxony Basin.



Figure 2 – A: Averaged mid-depth porosity variability of the post-Rupelian overburden, B: Distribution of the average sand content in the post-Rupelian overburden, C: Depth of the top of the Rupel Formation, also representing the thickness of the overburden (m), D. Spatial distribution of geothermal flux values(W/m²). The flux is significantly higher in the (north)west than the (south)east of the country





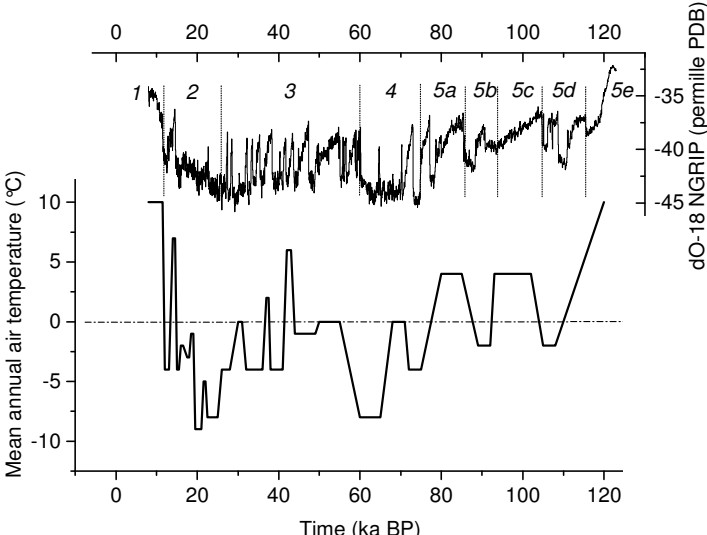

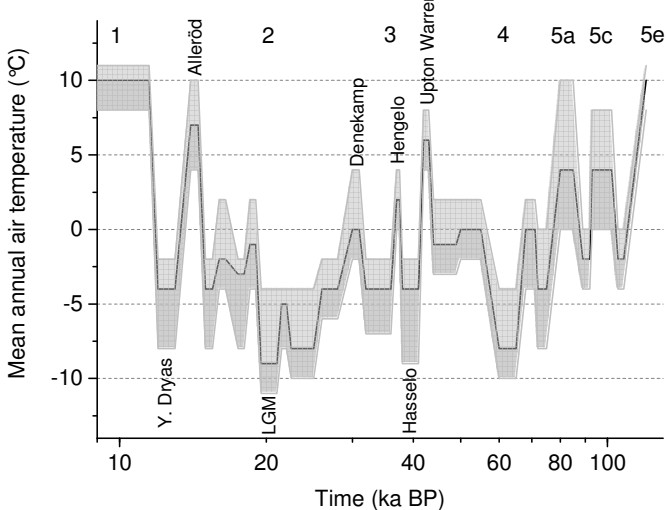

Figure 3 – Top: Best estimate temperature evolution for the Weichselian glaciation, which is taken as an analogue for a
future glacial climate in permafrost calculations. The curve is based on data from van Gijssel (1995), Huijzer and
Vandenberghe (1998), Renssen and Vandenberghe (2003), Busschers et al. (2007) and Buylaert et al. (2008). Marine isotope
stages (numbering 1 to 5e) are taken from Busschers et al. (2007) and references therein. The oxygen isotope curve is
reproduced from NGRIP (2004) data. Bottom: Upper and lower bound values for stochastic permafrost calculations on a
logarithmic timescale. MIS 5e is equivalent to the Eemian interglacial, while MIS 1 corresponds to the Holocene (present
interglacial). Y. Dryas = Younger Dryas.





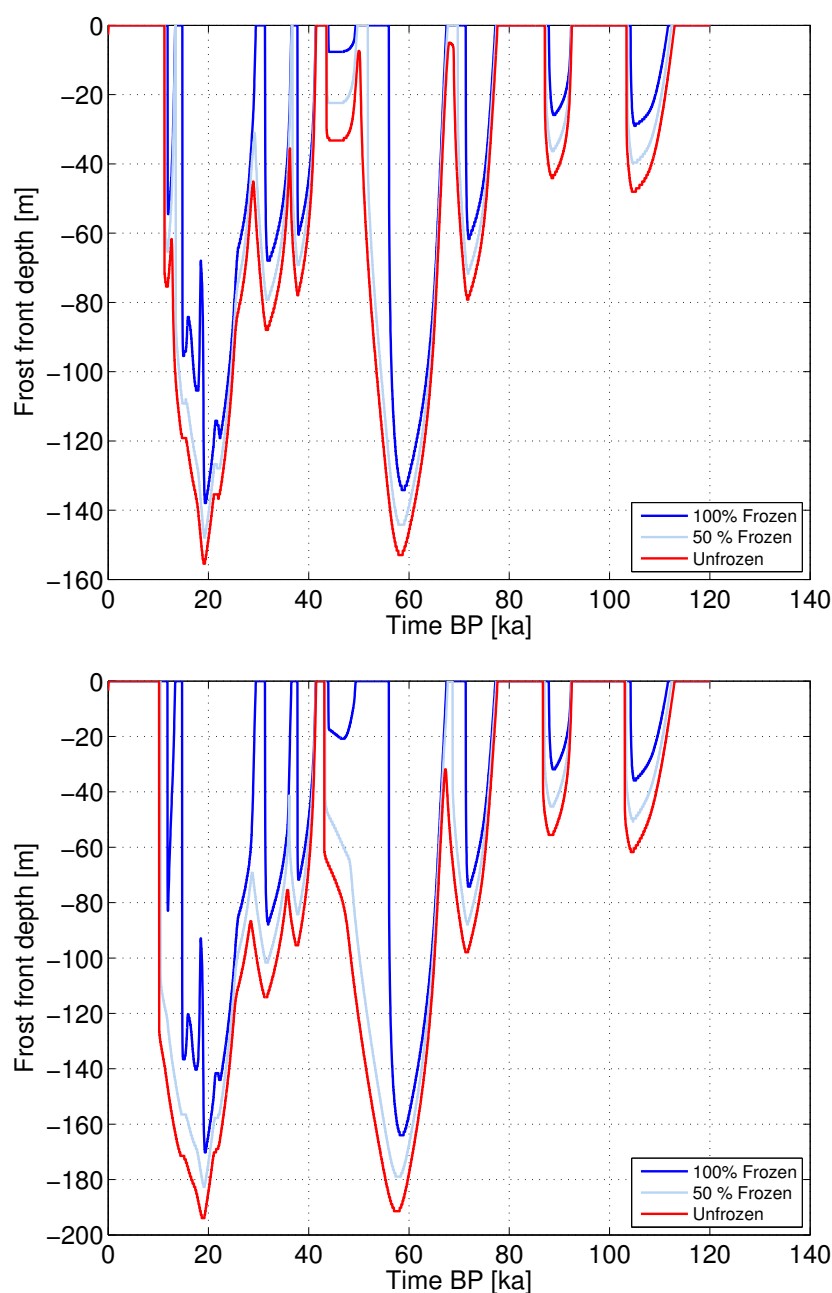

Figure 4: Top: Permafrost progradation during a simulation of the Weichselian glaciation cycle for the FRP polygon Bottom: Idem, for the LBH polygon.



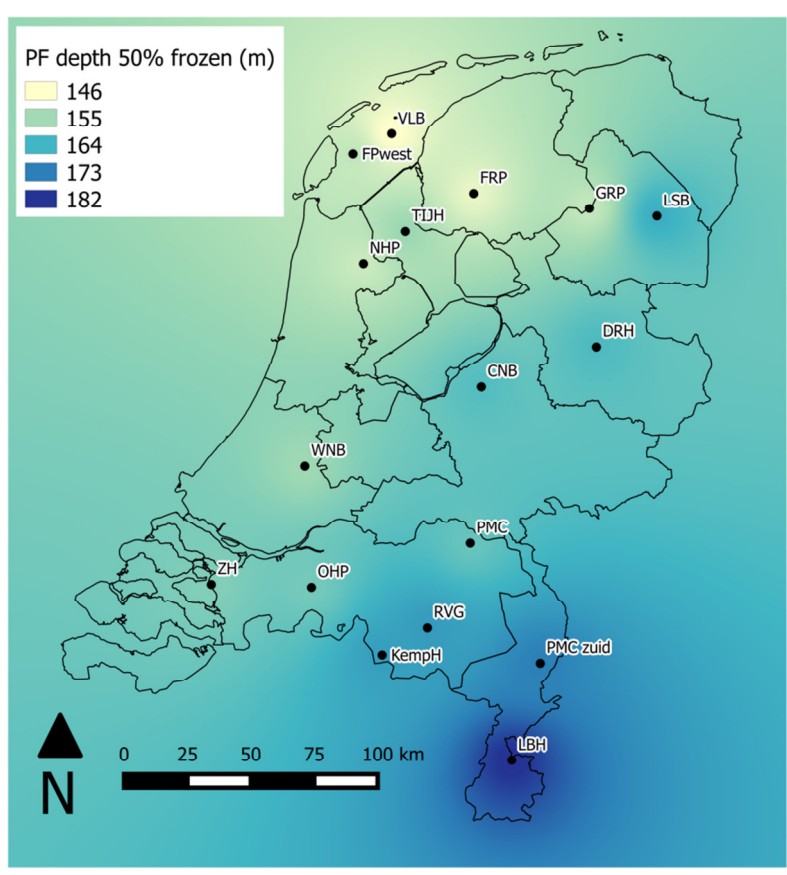

Figure 5 – Interpolated best estimate maximum permafrost depth map for the -0.25°C isotherm. Fifty percent of the pore water along this contour map is in the liquid phase.



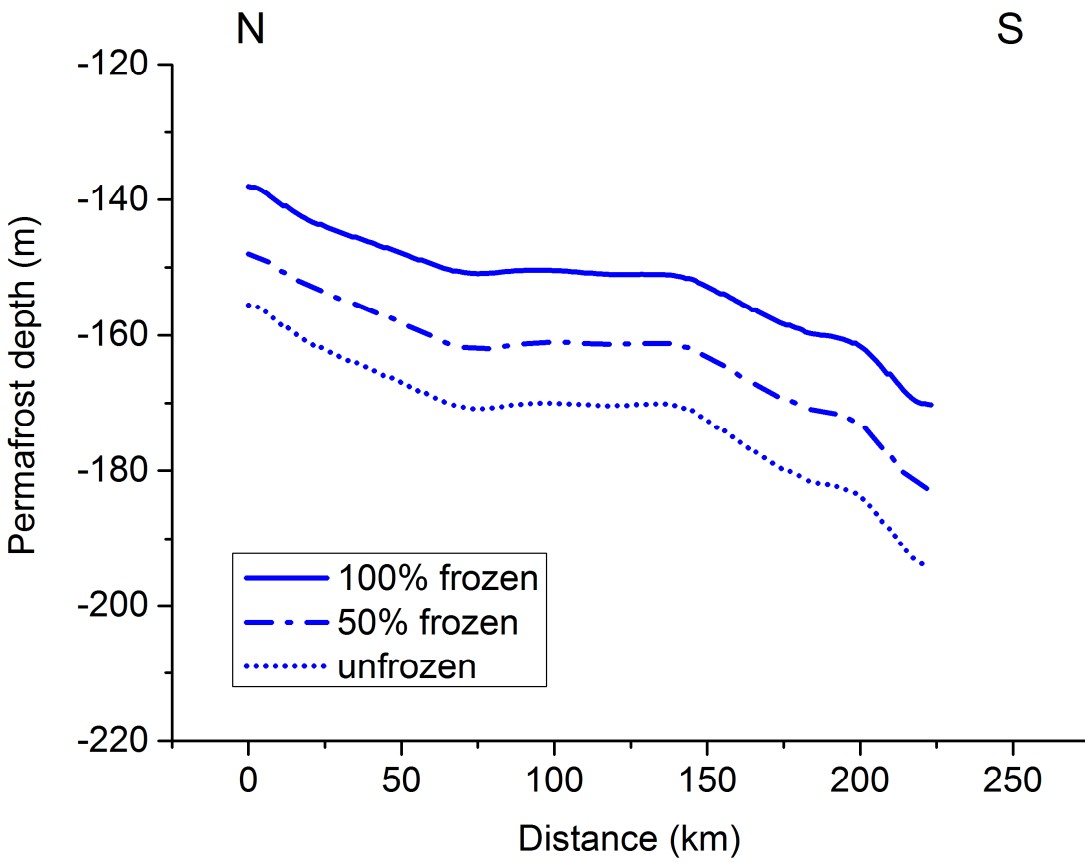

Figure 6 – Permafrost depth for different freezing states along a N-S transect from polygon centre FRP in the north to LBH in the south.





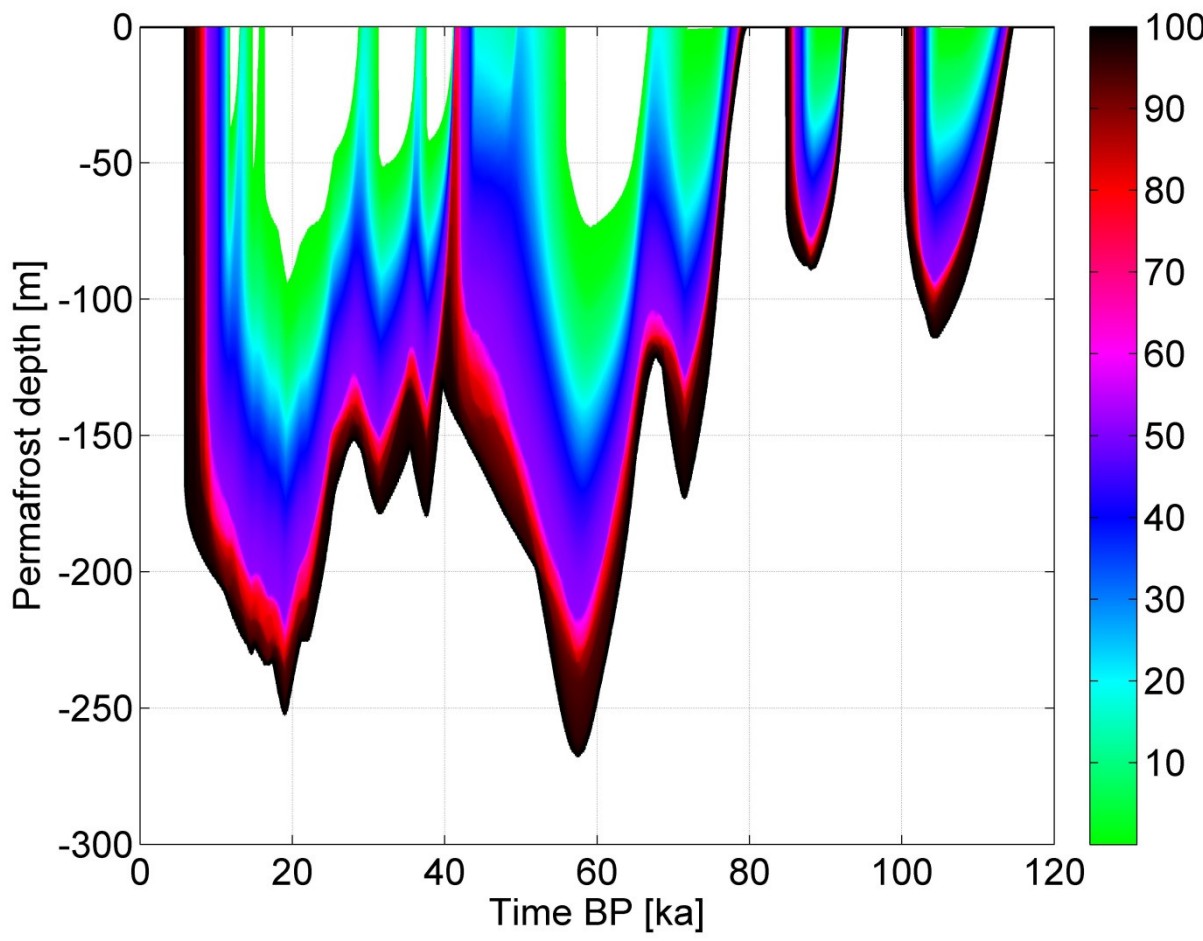

Figure 7: All percentiles of permafrost front penetration during a stochastic nation-wide simulation of the Weichselian
glaciation.



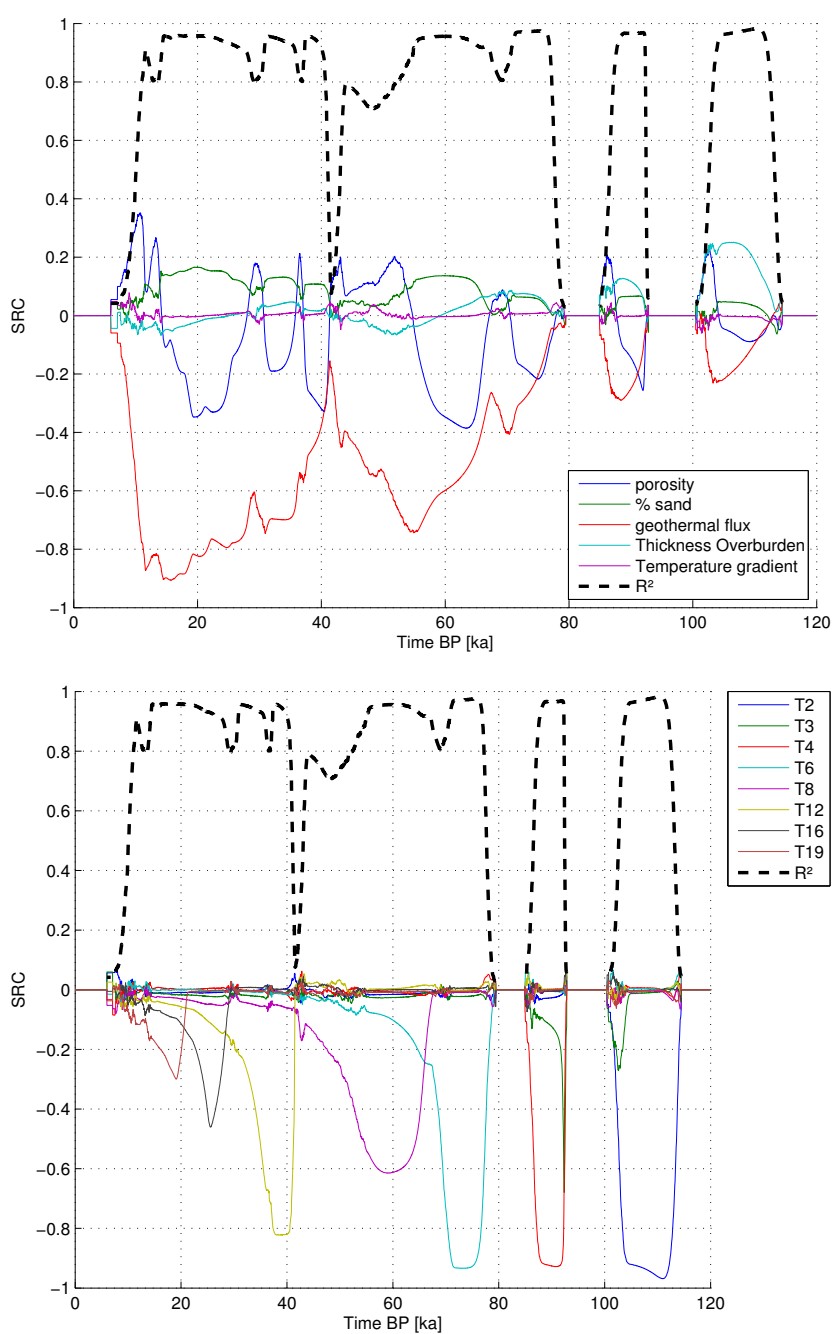

Figure 8: SRCs as a function of time for a global sensitivity sensitivity study of permafrost progradation during a Weichselian glaciation (top: physical parameters, bottom: selected set of temperature parameters).



Table 1: Properties of the different components of the subsoil.

| Parameter | Water | Ice | Clay Matrix | Sand Matrix |
|---|---|---|---|---|
| **Density [kg/m³]** | 997 | 918 | 2803 | 2358 |
| **Porosity** | - | - | 0.39 | Variable |
| **Specific Heat [J/(kg K)]** | 4185 | 1835 | 820 | 800 |
| **Thermal conductivity [W/(m K)]** | 0.54 | 2.37 | 1.98* | 3.00 |

*This value has been chosen so the effective thermal conductivity equals 1.31 J/(kg K), which is the vertical thermal conductivity of Boom Clay obtained during the ATLAS study (Cheng et al., 2010).



Table 2 – Parameters and associated ranges used in the global UA and SA. Additionally to the ones shown in the table, another 26 variables (T1 to T26) are used to control the magnitude of the various temperature plateaus during the Weichselian temperature cycle (Fig. 3), ranging from -11°C for the lowest, and +11°C for the highest.

| Parameter | Minimum | Maximum | Mode |
|---|---|---|---|
| Porosity [-] | 0,2 | 0,7 | 0,45 |
| Fraction of sand [-] | 0,1 | 1 | 0,75 |
| Geothermal flux [W/m²] | 0,033 | 0,115 | 0,060 |
| Overburden thickness [m] | 20 | 1500 | 500 |
| Initial Temperature gradient [K/m] | 0,022 | 0,033 | 0,028 |

