# Peer review of "Weichselian permafrost depth in the Netherlands: a comprehensive uncertainty and sensitivity analysis"

_The Cryosphere, 2016_

## Referee Comment (RC1) · Anonymous Referee #1 · 6 Jun 2016

Govaerts et al. conduct a study of potential future permafrost aggradation in the Netherlands in the context of nuclear waste repository installation. They go beyond most previously published studies in this area of research by conducting a sensitivity analysis to a number of factors including subsurface parameters as well as the forcing climate conditions. In general, this is a fairly clear, succinct paper that should garner some interest in The Cryosphere. I only have a few very minor comments for the authors to consider. Since these are all minor, I haven't structured the comments.

P1, L10, delete 'being'

P1, L14, insert 'the' before 'engineered'

[Figure]

P2, L3 Here and elsewhere, the authors interchange 'permafrost' with 'frozen ground'. Permafrost is only defined based on temperature (cryotic conditions) and does not necessarily imply the ground is frozen. This should be rewritten. Also in section 3.1 (first paragraph), the authors use -0.25C as their permafrost boundary. This makes no sense. By definition, the 0C isotherm is the permafrost boundary. The -0.25C level might be an indicator of frozen ground.

The first two sentences in the introduction need citations.

P2, L25, insert 'the' before 'engineered'

P2, L25-27, Kurylyk et al. (2014) review how permafrost separates surficial and deep subsurface water flow systems. They also discuss the role of advection in terms of the interactions between permafrost and climate. This would be useful in the authors' discussion on these topics later.

P3, L3 and elsewhere, is 'OPERA-project' an acronym? If so, define.

Equation (3). I am not used to seeing two derivatives (change in moisture content with temperature) in the effective heat capacity function including freeze-thaw. How do the authors reconcile this equation with Equation (14) in Kurylyk et al.?

Section 2, The authors don't really present the soil freezing curve (relationship between temperature and unfrozen water). They state on P5, L31 that they use a smooth Heaviside function. Heaviside function is not smooth, so this seems contradictory. Is it a linear function between 0 and -0.5C? If so, they should state that. If not, they should present the equation for it.

P5, L5 and L8, heat capacity (in this paper) is volume based, so why do the authors present it in mass-based terms (J/(kg K))

P5, L19, it is a bit silly to say that the thermal properties of the geologic material agree with the values chosen for similar material in past studies within the same order of magnitude. Surely one can be more precise than that given that thermal conductivities

of ALL geologic material only vary by about one order of magnitude.

P5, L22, those are not units of thermal conductivity

P6, L31, I'm confused by the term 'unit thickness' followed by 250x250. I guess the authors mean the geologic unit, but in modeling, unit thickness usually means a thickness of 1.

P7, L2, change 'which dimensions range' to 'with dimensions ranging'

P7, L12, porosity also affects the latent heat, not just the bulk thermal properties

P7, L22, perhaps it is stated elsewhere and I missed it, but what is the lower boundary condition? Is it specified flux or specified temperature? The authors could add a figure showing their domain and boundary conditions. I think that would be helpful. What is the time step size?

P9, L15, these are called 'thawing and freezing n-factors' not just 'thawing and freezing factors'

P9, L19, see beste type

P9, L30, insert 'of' after 'average'

P11, L10 delete 'a'

P11, L16, delete 'e.g.'

P12, L2 should 'en' be 'and'?

P15, L17, Was Govaerts et al. (2011) only for one site? If so, this should be stated here. If not, the distinguishing factors between the present study and the Govaerts et al. (2011) study should be more clearly outlined in the introduction.

P16, L4-7, wouldn't it make sense for the authors to include the fact that they ignored surface glaciation as another one of their 'conservative assumptions' that they list in two other locations? Maybe that's not relevant for the Netherlands

In general, I would assume the authors are familiar with the depths of the proposed nuclear waste repositories in the Netherlands. How do those design depths compare with the depths of maximum simulated permafrost? Surely this would be of interest to most readers.

Figure 4, why do the authors present results for these two specific polygons?

Figure 5, why do the authors show a binned color scale? Shouldn't this be a gradient color scale, or do the authors actually bin their results?

Figure 6, I think it would be advantageous for the authors to present the location of the transect for this figure in Figure 1

Figure 7, does this only show the maximum permafrost depth across the nation at any point in time, or is the spatial variability in permafrost included in the percentile calculations?

Figure 8, I'm confused by what T2-T19 refer to.

Table 2, is the porosity in Table 2 only the porosity of sand (Table 2 implies this). There are commas where there should be periods for the decimals.

---

## Referee Comment (RC2) · D. Kitover (Referee) · 14 Jun 2016

The discussion paper "Weichselian permafrost depth in lowland Europe: a comprehensive uncertainty and sensitivity analysis by Govaerts et al. 2016 explores potential permafrost depth give a range of uncertainty in The Netherlands. This investigation is compelled by the underlying possibility for future storage of nuclear waste.

General Comments:

Overall, this work is understandable and presents a methodological approach to assign estimates given a large range of uncertainty. However, there are some areas which seems to give superfluous information and unnecessary details while some other sec-

tions require more explanation and greater detail.

I think it is important to include more detail on the parameters of the model and sub-surface characteristics and how they vary. At one point, thermal conductivity is said to be averaged. However, from the model description they seem to change as a function of frozen/unfrozen. It is not clear what varies spatially, either from polygon to polygon, or vertically through the 1-d mesh. The porosity is mentioned plenty of times but is it stated that the model assumes full saturation and that the porosity equals water content? If this study is to be added into the inventory of past permafrost estimates, the understanding of the model construction, assumptions, initialization, and parameter settings should be very clear. This allows future research to make fair comparisons.

The description of the uncertainty analysis on page 10 should be condensed. It distracts the reader.

Specific Comments:

P2, L1-2: This first sentence needs a reference.

P2, L11: "periglacial conditions will reappear". Reappear where? At current state, periglacial conditions are existing already. Please be more specific at this kind of suggestion.

P5, L32: What do you mean by "conservatively" neglected?

P6, L13: This is the point at which the vertical description of the model is presented and it is overlooked. What is the depth and how does the mesh reflect the varying subsurface characteristics? This is confusing with your description on P6, L31. How deep is the lower boundary?

P6, L22: "The input temperature is held constant for the entire country". Do you mean for initialization or a spin-uo?. Surely, the temperature forcing changes temporally and spatially so what is this sentence describing? I suppose the intial setup? And what is the input temperature?

P6, L30 to P7, L2: I don't understand these dimensions and the following descriptions. This section should be described with improved clarity.

P7, L14-15: REGIS model should have a reference.

P8, L3: you may want to replace "input" here, and in other spots in manuscript. with the term "forcing".

P9, L13: ..as soil input data.." is vague. Maybe write surface temperature data or land temperature forcing..

P9, L17: "shielding effect of snow and temperature" does not make sense

P9, L30: "average thermal conductivity values". Does this mean there is only one thermal conductivity value per polygon or just for the purpose of calculating a thermal gradient?

P12, L3-5: The T1 to T26 are simply time intervals or snapshots of the transient simulation? Is this a variable you control for? A table should be provided that relates the T variables with their time interval.

P14, L9: You state that a higher parameter will cause a larger permafrost depth but how is this possible if for example, the geothermal heat flux parameter is high, which in turn would cause less permafrost to develop ?

P14, L1-9: If you state earlier that you use SRC for this study (P12, L20), why are you discussing PCC?

P14, L16: How can you say that geothermal heat flux is the main driving force of degradation? Is it not the temperature forcing at the surface? Is this accounted for by the R2? If you find that the geothermal heat flux accounts strongly for permafrost warming MORE than the surface temperatures, this is an important finding that should be clarified and explained.

P15, L9-12: This is an important point but should be referenced by earlier studies which

discuss how surface forcings penetrate deeper into soil and the time frame associated with the lag.

P15, L15 and further: Although you are comparing your estimates at 50% frozen this is probably not the case with the other studies, which assume 100% frozen and then this depends on what their freezing algorithm was, what temperature etc.

Almost all the figure and/or figure captions should be improved: Figure 1: Make sure all 17 polygons are represented and listed - my count was off. Figure 2: "mid-depth" porosity, is this half-way ? Figure 4: Consider making y-axis scale same for both figures to more easily compare the two sites. Figure 6: X-axis "Distance" from what? Maybe put a little insert to illustrate the distance. Figure 7: This figure caption does not explain what the percentile is of? The secondary x-axis should have a unit associated with it (% I think). Figure 8: Both figures should align even on top of each other for more illustrative comparison. Permafrost thickness should be also illustrated with the curved, perhaps in a third figure below.

Technical corrections:

Make sure at first mention of acronyms that they are defined. For example. . .DGM

P1 L10 remove 'being'

P3 L10 replace 'so-called' with 'kind of'

P3 L24 replace "the thorough" with "a more thorough"

P5 L20 – L30: Check consistent use of units Figure 1: Are all the structural elements (17) listed in the figure caption and on the map? Are the names consistently referenced?

P8, L33: Remove "including"

P8, L33: should read ". . .future glacial colder than.."

P8, L20: remove "conservative" and write "..the model neglects subsurface hydrology such as the vadose zone and groundwater flow. During very cold..."

P11, L4: remove "exhaustively"

P11, L14: correct as such "tables of collected results produced by Matlab can then be directly analysed to calculate and plot the.."

P11, L16: Would sound better to begin sentence "Matlab was then used again to compute"

P11, L 20: Are you just being redundant from line 17 by again stating you performed 1000 realizations?

P12, L2: extra period

P12, L5: comma before which

P12, L12: remove "(and other local SA methods)"

P12, L29: should be written "..are shown as a function of time.."

P14, L21: remove "away"

P16, L8: don't end sentence with a preposition such as "to"

P17, L3: "error" is not a good word here. Consider a less negative word such as discrepancy or unknowns

---

## Author Comment (AC1) · 22 Aug 2016

Govaerts et al. conduct a study of potential future permafrost aggradation in the Netherlands in the context of nuclear waste repository installation. They go beyond most previously published studies in this area of research by conducting a sensitivity analysis to a number of factors including subsurface parameters as well as the forcing climate conditions. In general, this is a fairly clear, succinct paper that should garner some interest in The Cryosphere. I only have a few very minor comments for the authors to consider. Since these are all minor, I haven't structured the comments.

The authors wish to thank the reviewer for his/her time and positive remarks on the manuscript. We are also thankfull for the relevant remarks about the manuscript, which

in our opinion improved the paper.

P2, L3 Here and elsewhere, the authors interchange 'permafrost' with 'frozen ground'. Permafrost is only defined based on temperature (cryotic conditions) and does not necessarily imply the ground is frozen. This should be rewritten. Also in section 3.1 (first paragraph), the authors use -0.25C as their permafrost boundary. This makes no sense. By definition, the 0C isotherm is the permafrost boundary. The -0.25C level might be an indicator of frozen ground. In appendix A of Govaerts et al. (2011) which can be found on http://publications.sckcen.be/dspace/handle/10038/7377 , we demonstrated that the evolution of the the temperature profiles throughout the simulation time are not very sensitive to the choice of the width of the liquid-solid interval interval. However, concerning the safety of a radwaste disposal facility, the penetration depth of the fully frozen front is of more relevance than the temperature. On the other hand, the positions of the 0 and 100 % frozen isolines are severely sensitive to this width, and the exact value is uncertain as it can range from 0.5 to 2 °C depending on the material type (Noetzli & Gruber, 2009). Therefore, the choice for the -0.25°C as permafrost indicator was made (i.e. the center temperature of the 0°C to -0.5°C freezing interval which coincides with the 50% frozen isoline) as the main output of interest in this study, in order to present our results in a robust manner, independent from modelling assumptions. The 50% frozen isolines serves as a pessimistic indicator for the fully frozen front, including a safety margin. (the previous part has been added to the manuscript) In a first version of this manuscript, the freezing interval was chosen as 0.5°C to -0.5°C, and the 0°C – isotherm was used as the main output. However, the editor could not agree with the fact that water would start to freeze at temperatures above 0°C. Therefore, we have changed the offset and the width of the liquid-to-solid interval, now 0°C to -0.5°C and have rerun all the simulations, for the nationwide best estimate analysis and the stochastic runs. The new results where then added to the manuscript, but the differences with the previous results were rather subtle (see figure below). As such, no large changes were made in the results section, except for the figures, who were replaced with the latest results.

P2, L25-27, Kurylyk et al. (2014) review how permafrost separates surficial and deep subsurface water flow systems. They also discuss the role of advection in terms of the interactions between permafrost and climate. This would be useful in the authors' discussion on these topics later. Thank you for your suggestion, we have made a reference to this work. Equation (3). I am not used to seeing two derivatives (change in moisture content with temperature) in the effective heat capacity function including freeze-thaw. How do the authors reconcile this equation with Equation (14) in Kurylyk et al.? It must be noted that the two derivatives in equation 3 do not represent the change in absolute moisture content, but the change in the fluid fraction with respect to the total porosity. Inserting the relations of equation (2) in here and neglecting the difference in density between water and ice, will transform this equation into one comparable to Eq. 14 in Kurylyk et al., (2014) with only one derivative.

Section 2, The authors don't really present the soil freezing curve (relationship between temperature and unfrozen water). They state on P5, L31 that they use a smooth-Heaviside function. Heaviside function is not smooth, so this seems contradictory. Is it a linear function between 0 and -0.5C? If so, they should state that. If not, they should present the equation for it. 'Smoothed Heavised function' has been replaced by 'a fifth order S-shaped polynomial form (available in COMSOL as the inbuilt function flc2hs). The polynomial form is a smoothed Heaviside function with continuous second derivative without overshoot and takes on a value between 0 and 1.' P5, L5 and L8, heat capacity (in this paper) is volume based, so why do the authors present it in mass-based terms (J/(kg K)). Indeed, the Ceq of equation 3 is volume based. However, the values heat capacities of unfrozen and frozen Boom clay given here are mass based as they have been obtained by dividing the equivalent volume based heat capacity with the bulk density. This was done in order to make a straightforward comparison to values used by other authors. We have added a little clarification in this paragraph to avoid this confusion. P5, L19, it is a bit silly to say that the thermal properties of the geologic material agree with the values chosen for similar material in past studies within the same order of magnitude. Surely one can be more precise than that given

that thermal conductivities of ALL geologic material only vary by about one order of magnitude. 'Orders of magnitude' has been replaced by 'range'. I'm confused by the term 'unit thickness' followed by 250x250. I guess the authors mean the geologic unit, but in modeling, unit thickness usually means a thickness of 1. The sentence has been rearranged: "For each unit, vertical gridcells of 250x250m surface with a height equal to the thickness of the unit were constructed." P7, L12, porosity also affects the latent heat, not just the bulk thermal properties The following changes have been made for completeness: "Porosity is directly linked with water content as full saturation is assumed and thus thermal conductivity and the equivalent heat capacity of the soil. (see Table 1 and Equation (3))." P7, L22, perhaps it is stated elsewhere and I missed it, but what is the lower boundary condition? Is it specified flux or specified temperature? The authors could add a figure showing their domain and boundary conditions. I think that would be helpful. What is the time step size? Information about the bottom boundary condition is given in section 2.4. A separate paragraph concerning the model domain, boundary condition and computational settings has been added (2.5). P15, L17, Was Govaerts et al. (2011) only for one site? If so, this should be stated here. If not, the distinguishing factors between the present study and the Govaerts et al. (2011) study should be more clearly outlined in the introduction. "As such, the work performed in Govaerts et al. (2011), which was done for one potential site in the framework of the Belgian research programme on High-level waste disposal, is taken a few steps further." has been added to the introduction. P16, L4-7, wouldn't it make sense for the authors to include the fact that they ignored surface glaciation as another one of their 'conservative assumptions' that they list in two other locations? Maybe that's not relevant for the Netherlands. We ignored this as there were no ice sheets in the Netherlands during the Weichselian. In general, I would assume the authors are familiar with the depths of the proposed nuclear waste repositories in the Netherlands. How do those design depths compare with the depths of maximum simulated permafrost? Surely this would be of interest to most readers. "Note that in the OPERA-project the long term safety of a generic repository in the Boom Clay at a generic depth of 500 m

will be assessed (Verhoef and Schröder, 2011)." Has been added to the discussion. Figure 4, why do the authors present results for these two specific polygons? "These two polygons (FRP and LBH) are at resp. the low and the high end of the resulting permafrost depths." Has been added to the caption of the figures Figure 5, why do the authors show a binned color scale? Shouldn't this be a gradient color scale, or do the authors actually bin their results? The results are not binned. The QGIS software does not allow to create a gradient color scale. Figure 6, I think it would be advantageous for the authors to present the location of the transect for this figure in Figure 1. This has been done. Figure 7, does this only show the maximum permafrost depth across the nation at any point in time, or is the spatial variability in permafrost included in the percentile calculations? This figure indeed shows the percentiles maximum permafrost depths as a function of time of 1000 simulations. However, spatial variability is implicitly included as the parameter ranges include this uncertainty. Figure 8, I'm confused by what T2-T19 refer to This is explained in the paragraph 2.6.3: "T1 to T26 are variables which are used to control the magnitude of the various temperature plateaus during the Weichselian temperature cycle. This allows to account for the actual parameter uncertainty on the temperature as well as the nation-wide spatial parameter variability. "

Table 2, is the porosity in Table 2 only the porosity of sand (Table 2 implies this). There are commas where there should be periods for the decimals. Yes, the porosity of the Clay material is kept constant, as the variability is much lower. We have adapted the table.

Please also note the supplement to this comment:
http://www.the-cryosphere-discuss.net/tc-2016-54/tc-2016-54-AC1-supplement.pdf
* * *
[Figure]

[Figure]

**Fig. 1.** Results of the uncertainty analysis. Top: previous version (freezing interval 0.5°C to -0.5°), bottom: after first revision (freezing interval 0°C to -0.5°). Differences on the extreme percentiles ar

---

## Author Comment (AC2) · 22 Aug 2016

General Comments: *Overall, this work is understandable and presents a methodological approach to assign estimates given a large range of uncertainty. However, there are some areas which seems to give superfluous information and unnecessary details while some other sections require more explanation and greater detail.*

We would like to thank reviewer 2 for her remarks; we have tried to address them as good as possible. We believe this has improved the quality of the manuscript.

*I think it is important to include more detail on the parameters of the model and sub-surface characteristics and how they vary. At one point, thermal conductivity is said to

be averaged. However, from the model description they seem to change as a function of frozen/unfrozen. It is not clear what varies spatially, either from polygon to polygon, or vertically through the 1-d mesh. *

We have tried to make this more clear. See below in the specific comments.

*The porosity is mentioned plenty of times but is it stated that the model assumes full saturation and that the porosity equals water content? *

We state that vadose zone is not taken into account, which implies full saturation. We have stated it more explicity now.

*If this study is to be added into the inventory of past permafrost estimates, the understanding of the model construction, assumptions, initialization, and parameter settings should be very clear. This allows future research to make fair comparisons. *

We have tried to do this by adding more detail on the model domain and parameters in a new paragraph (2.5).

*The description of the uncertainty analysis on page 10 should be condensed. It distracts the reader.*

As acknowledged by the editor and reviewer 1, the uncertainty and sensitivity analysis is the main asset of this paper. The description on page 10 is necessary to frame the need for uncertainty analysis within the context of radioactive waste disposal. Also, we believe that the description of these techniques is already concise and does not go too much into mathematical details. We prefer to leave it as it is.

Specific Comments: *P2, L1-2: This first sentence needs a reference.

This has been done.

*P2, L11: "periglacial conditions will reappear". Reappear where? At current state, periglacial conditions are existing already. Please be more specific at this kind of suggestion.

[Figure]

We have added some clarifications at this point.

*P5, L32: What do you mean by "conservatively" neglected?

Conservative means that including these effects would most likely decrease permafrost depths. As such, a safety margin on the results is implicitly included.

*P6, L13: This is the point at which the vertical description of the model is presented and it is overlooked. What is the depth and how does the mesh reflect the varying subsurface characteristics? This is confusing with your description on P6, L31. How deep is the lower boundary?

We have added a paragraph which treats all this in detail (section 2.5) including a figure which should help to understand how the model domain is set up.

*P6, L22: "The input temperature is held constant for the entire country". Do you mean for initialization or a spin-uo?. Surely, the temperature forcing changes temporally and spatially so what is this sentence describing? I suppose the intial setup?

"The input temperature is allowed to change temporally, but held uniform spatially for a given time step." This has been added.

*And what is the input temperature?

We have replaced input temperature with 'forcing' temperature, as suggested.

*P6, L30 to P7, L2: I don't understand these dimensions and the following descriptions. This section should be described with improved clarity.

We have restructured this sentence in order to make it more clear.

*P7, L14-15: REGIS model should have a reference.

A reference to the REGIS II model has been added.

*P8, L3: you may want to replace "input" here, and in other spots in manuscript. With the term "forcing".

This has been done.

\*P9, L13: ..as soil input data..” is vague. Maybe write surface temperature data or land temperature forcing.

This has been done.

\*P9, L17: “shielding effect of snow and temperature” does not make sense

...and temperature has been removed.

\*P9, L30: “average thermal conductivity values”. Does this mean there is only one thermal conductivity value per polygon or just for the purpose of calculating a thermal gradient?

Yes, the thermal conductivity of the Rupelian Clay overburden is averaged across the entire depth. This is used in the simulations.

\*P12, L3-5: The T1 to T26 are simply time intervals or snapshots of the transient simulation? Is this a variable you control for? A table should be provided that relates the T variables with their time interval.

“T1 to T26 are variables which are used to control the magnitude of the various temperature plateaus during the Weichselian temperature cycle. This allows to account for the actual parameter uncertainty on the temperature as well as the nation-wide spatial parameter variability.” Is stated in section 2.6.3. They have been added to table 2.

\*P14, L9: You state that a higher parameter will cause a larger permafrost depth but how is this possible if for example, the geothermal heat flux parameter is high, which in turn would cause less permafrost to develop ?

“A positive correlation coefficient (SRC/PCC) means that a higher value of the parameter will cause a larger permafrost depth and vice versa. “. So if an increase of the parameter value increases the permafrost depth, the sensitivity coefficient is positive. If an increase of the parameter value decreases the permafrost depth, the sensitivity

coefficient is negative.

*P14, L1-9: If you state earlier that you use SRC for this study (P12, L20), why are you discussing PCC?

This was a mistake, it has been corrected.

*P14, L16: How can you say that geothermal heat flux is the main driving force of degradation? Is it not the temperature forcing at the surface? Is this accounted for by the R2? If you find that the geothermal heat flux accounts strongly for permafrost warming MORE than the surface temperatures, this is an important finding that should be clarified and explained.

It can be seen in figure 8, that during the timeframe of decreasing permafrost depth (for instance 10 − 20 ka BP, 40 − 50 ka BP) the SRC of geothermal flux becomes larger than those of the temperatures relevant for that time period. It must be noted that the SRCs account for the sensitivity of the permafrost depth. So it makes sense that the geothermal flux will act more severely at the base of the permafrost rather than the surface temperature, which will force the melting at the top of the domain. This is particularly true when the permafrost front has reached greater depths. We have changed the sentence in the following way in order to be more specific about the influence of the geothermal flux: "However, when the surface temperature again rises and the permafrost starts to degrade, the geothermal flux acts as the main driving force of the melting process at the base of the permafrost, resulting in a decrease of the permafrost depth."

*P15, L9-12: This is an important point but should be referenced by earlier studies which discuss how surface forcings penetrate deeper into soil and the time frame associated with the lag.

We have added a reference.

*P15, L15 and further: Although you are comparing your estimates at 50% frozen this

is probably not the case with the other studies, which assume 100% frozen and then this depends on what their freezing algorithm was, what temperature etc.

Please refer to our answer to reviewer 1 concerning the choice of the permafrost depth indicator. We do not wish to use the 100% frozen isoline as its position is highly sensitive to the choice of the solid-liquid interval width. A lot of studies use the 0°C isotherm as an indicator, which would allow to compare with our results (50% frozen $\sim$ -0.25°C ).

*Almost all the figure and/or figure captions should be improved: Figure 1: Make sure all 17 polygons are represented and listed - my count was off.

This has been adapted.

*Figure 2: "mid-depth" porosity, is this half-way ?

Yes.

*Figure 4: Consider making y-axis scale same for both figures to more easily compare the two sites.

This has been done.

*Figure 6: X-axis "Distance" from what? Maybe put a little insert to illustrate the distance.

See Figure 1 for location of the profile.

*Figure 7: This figure caption does not explain what the percentile is of? The secondary x-axis should have a unit associated with it (% I think).

Percentiles do not have a unit. 'Depth' has been added to the caption.

*Figure 8: Both figures should align even on top of each other for more illustrative comparison. Permafrost thickness should be also illustrated with the curved, perhaps in a third figure below

We have aligned these figures. Figure 8 can be easily compared with figure 5. We think that adding more curves to these figures would compromise the readibility of these - already busy - figures.

Sincerely, J. Govaerts, K. Beerten, J. Ten Veen.

Please also note the supplement to this comment:
http://www.the-cryosphere-discuss.net/tc-2016-54/tc-2016-54-AC2-supplement.pdf

[Figure]

**Supplement:**

[revised manuscript text omitted]
. Concerning the safety of a geological radwaste disposal facility, the penetration depth of the fully frozen front is of more relevance than the temperature. However, the positions of the 0 and 100 % frozen isolines are severely sensitive to this width, and the exact value is uncertain as it can range from 0.5 to 2 °C

15 depending on the material type (Noetzli & Gruber, 2009). Therefore, the choice for the -0.25°C as permafrost indicator was made (i.e. the center temperature of the 0°C to -0.5°C freezing interval which coincides with the 50% frozen isoline) as the main output of interest in this study, in order to present our results in a robust manner, independent from modelling assumptions. The 50% frozen isolines serve as a pessimistic indicator for the fully frozen front, including a safety margin.

[revised manuscript text omitted]

---

## Author Response (AR2)

*Dear authors,*

*Thank you for your revised manuscript and your rebuttal letter. I have carefully read the revised version and find it suitable for publication after minor (but many) revisions as outlined below. Please implement these changes carefully and provide a justification where you disagree with my suggestions. To speed up the further process, please also provide a highlighted version of the revised manuscript so your changes can better be tracked.*

*Kind regards,*
*Stephan Gruber*

Thank you, we will try to address your questions and suggestions in the following.

*Title: Weichselian permafrost depth in the Netherlands: an uncertainty and sensitivity analysis (the Po Basin would also be lowland Europe...)*

We agree. The title has been adapted.

*MAJOR COMMENTS*
*Why is the "fully frozen front" the relevant metric (P6L13)? I have several issues with this: (a) no justification is given, (b) down to -40ºC you will not find 'fully' frozen soil/rock and thus this only exists in your current parameterization – with all the inherent arbitrariness, (c) one would assume that the presence of any amount of ice would be relevant as this begins to affect hydraulic permeability and material structure. If you have a good reason to stick to 'fully frozen', please justify this well. If not, I suggest to us the definition of permafrost (<0ºC) as this is (a) conservative and (b) provides a definite distinction of ground that can/cannot have ice. See next comment.*

*You give a valid permafrost definition, and at the same time you use a conflicting and arbitrary definition of -0.25ºC as an indicator (P6L16, P13L8–10, P16L14, P17L31). This is arbitrary and confusing.*

The 'complete' freezing of the soil near the radio-active waste repository would be the most penalising situation for the integrity of the engineered components of the system (concrete elements, bentonite seals, etc...). Therefore, the frozen front is the most relevant metric is this study. However, we agree that the choice for the -0.25 °C isotherm (or 50% frozen), which is based on the consideration to use the center temperature of the freezing interval, seems arbitrary when not familiar with the radio-active waste disposal concept. We therefore agree to use the 0°C isotherm as the permafrost indicator, which is in line with the given permafrost definition and makes this study easier to compare with similar permafrost modelling studies. All necessary figures were adapted and permafrost depth increased with

about 10 m due to the change of metric. The results and conclusions of the sensitivity analysis were not influenced by this.

*Distinguish the description of material freezing behaviour and the parameterization used in your numberc scheme (P6L2/3 shows lack of understanding, as your liquidus/solidus parameterization of freezing may be interpreted of representing pore diameter and surface effects).*

We have added the following: "As such, in this model representation, the freezing process is determined only by the change in temperature. The dependency of the freezing point of water on pressure, salinity and other possible influencing factors is not taken into account in the present calculations."

*P4L25–28: Please reformulate to clarify that (a) you describe the behavior of your parameterization (in reality there will be unfrozen water below you solidus) and (b) that this describes the behavior of water in a porous medium (a mixture of substances and not a pure material), correspondingly the term used should be 'thaw' rather then 'melt'.*

We have changed the text into: "During cooling, solidus is that temperature at which most of the pore water of the soil is frozen. Between the solidus and liquidus temperatures, there will be a mixture of solid and liquid water phases within the soil matrix. Just below the liquidus temperature, there will be mostly liquid water phases."

All minor comments have been addressed in the marked version of the manuscript.

*Please check you references for consistency with respect to capitalization and quotation marks.*

This has been done.

Thank you,

Joan Govaerts

Koen Beerten

Johan Ten Veen

---

## Author Response (AR3)

**Weichselian permafrost depth in the Netherlands: a comprehensive uncertainty and sensitivity analysis**
**Reply to editor**

*Dear authors,*

*thank you for your rebuttal and your revised manuscript. Please check you text again, thoroughly, and resubmit.*

*In you rebuttal you write "We therefore agree to use the 0°C isotherm as the permafrost indicator" and still I find the old concept inconsistently interspersed in your manuscript, e.g., P15L26: "The best estimate permafrost depth values of 140-180 m (50% ice and 50% water)" or P17L7: "indicates that the permafrost front (50% ice and 50% water) would".*

*Kind regards,*
*Stephan Gruber*

Dear Editor,

We thank you for spotting these inconsistencies. We carefully went throught the manuscript, and have adapted the text in order to comply with the applied definition of permafrost depth. Changes are marked in green.

Thank you,

Joan Govaerts

Koen Beerten

Johan Ten Veen